# The Natural Course of *Chlamydia trachomatis*, *Neisseria gonorrhoeae, Trichomonas vaginalis,* and *Mycoplasma genitalium* in Pregnant and Post-Delivery Women in Pemba Island, Tanzania

**DOI:** 10.3390/microorganisms9061180

**Published:** 2021-05-30

**Authors:** Naomi C. A. Juliana, Abdulla Mbaruk Omar, Jolein Pleijster, Fahad Aftab, Nina B. Uijldert, Said M. Ali, Sander Ouburg, Sunil Sazawal, Servaas A. Morré, Saikat Deb, Elena Ambrosino

**Affiliations:** 1Department of Genetics and Cell Biology, Institute for Public Health Genomics (IPHG), Faculty of Health, Medicine & Life Sciences, Research School GROW (School for Oncology & Developmental Biology), University of Maastricht, 6229 ER Maastricht, The Netherlands; n.juliana@maastrichtuniversity.nl (N.C.A.J.); samorretravel@yahoo.co.uk (S.A.M.); 2Public Health Laboratory-Ivo de Carneri, Chake, Pemba Island, Tanzania; omarabdallambarouk@gmail.com (A.M.O.); fahadaftabjmi@gmail.com (F.A.); said@phlidc.org (S.M.A.); saikatdeb@gmail.com (S.D.); 3Laboratory of Immunogenetics, Department of Medical Microbiology and Infection Control, Amsterdam UMC, Vrije Universiteit, 1105 AZ Amsterdam, The Netherlands; pj.pleijster@amsterdamumc.nl (J.P.); n.uijldert@vumc.nl (N.B.U.); s.ouburg@amsterdamumc.nl (S.O.); 4Centre for Public Health Kinetics, New Delhi 110024, India; ssazawal@jhu.edu

**Keywords:** pregnancy, genital infections, Tanzania, sub-Saharan Africa, sexual and reproductive health, *Chlamydia trachomatis*, *Neisseria gonorrhoeae*, *Trichomonas vaginalis*, *Mycoplasma genitalium*

## Abstract

This study aimed to determine the persistence of *Chlamydia trachomatis* (CT), *Neisseria gonorrhoeae* (NG), *Trichomonas vaginalis* (TV) and *Mycoplasma genitalium* (MG) infections during pregnancy and after delivery in vaginal swabs of women from Pemba Island, Tanzania. In the context of an earlier biobanking effort, vaginal swabs were collected at two timepoints during pregnancy and once post-delivery. Detection of CT, NG, TV, and MG was performed by PCR using validated detection kits in samples from 441 pregnant women aged 16–48 years old. Among those, 202 samples were matched during pregnancy and 38 at the second timepoint of the pregnancy and post-delivery CT infection persistence during pregnancy was 100% (*n* = 11) after an average of eight weeks, that of TV infection 82% (*n* = 11) after ten weeks, and that of MG infection 75% (*n* = 4) after ten weeks. Post-delivery (after approximately 22 weeks) infection persistence was 100% for CT (*n* = 1) and 20% for TV (*n* = 5). NG was only detected at the last collection timepoint, its persistence rate could not be determined. These results show persistence and clearance of curable infections during and after pregnancy. Analysis of biobanked samples is a valuable approach in the investigation of the natural history of curable pathogens.

## 1. Introduction

*Chlamydia trachomatis*, *Neisseria gonorrhoeae, Trichomonas vaginalis* and *Mycoplasma genitalium* are common curable sexually transmitted infections (STIs) that are mostly asymptomatic and have a high burden in sub-Saharan Africa [1,2]. In 2016, the estimated prevalence among women between 15 and 49 years of age was 5% for *C. trachomatis* infection, 1.9% for *N. gonorrhoeae* infection and 12% for *T. vaginalis* infections in the African region [3]. For *M. genitalium,* no recent regional data for sub-Saharan Africa are available, but prevalence among reproductive-age women in South Africa has been reported between 1.6 and 11% [4]. These STIs can cause acute urogenital problems, such as inflammation of the cervix (cervicitis), but their often asymptomatic presentation (as is often the case for *C. trachomatis* and *N. gonorrhoeae* infections) is the reason why most cases remain undiagnosed, thus contributing to long-term sequelae such as pelvic inflammatory disease [3,5]. During pregnancy and delivery, vertical transmission might occur, possibly causing adverse neonatal outcomes [3]. In recent years, infections with these four pathogens have been, in turn, associated with, among others, premature delivery, low birthweight and neonatal sepsis [5,6,7,8,9,10,11,12]. Despite the availability of effective antibiotic treatments, the burden of STIs remains high, including among pregnant women, in most African countries [2,13].

Fortunately, natural protective immunity to these infections does occur in some women, but studies on their natural history, especially in pregnancy, are limited due to the standard care available once the pathogen is diagnosed [14,15,16]. Notwithstanding, one study observed that 44% of 140 women with asymptomatic *C. trachomatis* infection spontaneously cleared it over the course of 1 to 14 days during pregnancy [17]. Information about the natural history of *C. trachomatis*, *N. gonorrhoeae, T. vaginalis* and *M. genitalium* infections during pregnancy is scant. However, data exist on non-pregnant women [14,18,19,20,21]. A previous study on asymptomatic non-pregnant women showed the resolution rate of *C. trachomatis* to be 11–45% [14]. The follow-up intervals differed by weeks, months and even years [14]. For instance, *C. trachomatis* cleared in 45% of non-pregnant Dutch women in a year, and in 94% of non-pregnant Colombian women after a 4-year follow-up [22,23]. Studies showed that asymptomatic *N. gonorrhoeae* infection can persist as long as five months in non-pregnant women, and that close to 25% of patients clear the infection within that time period [21]. For *M. genitalium* infection in non-pregnant women, the median clearance time is about 2 months; one study showed that 93% of 119 women involved in the study cleared it within 12 months [18]. Finally, prediction studies suggest that the average duration of untreated *T. vaginalis* infection is 3 to 5 years in non-pregnant women [19,20]. The infection persistence rate differs between pregnant and non-pregnant women, possibly due to the difference in the immune response as a result of hormonal regulation [24,25]. For some infections, pregnant women are more susceptible, and at times more severely affected than non-pregnant women [25]. Inflammatory responses that are important to clear a pathogen can have negative consequences on the outcome of the pregnancy; pregnant women are biased towards an anti-inflammatory phenotype that is less aggressive towards disease pathogenesis, in a bid to protect viability of the pregnancy [24,25].

Designing longitudinal studies to investigate the persistence of curable infections is ethically challenging and for prospective studies unacceptable, but evidence on the natural history of infections is paramount, particularly among the most vulnerable hosts, as is the case for pregnant women; more so in communities with a high burden of STIs and related pregnancy complications, as is the case for Tanzania [7,26,27,28]. In the eastern part of the country, on the Island of Pemba, a biobanking effort was established in 2014, and follow-up studies have been set up to investigate contributors to maternal mortality and morbidity [29,30].

The present study retrospectively analyzed previously collected vaginal swabs to determine the persistence of *C. trachomatis*, *N. gonorrhoeae, T. vaginalis,* and *M. genitalium* infections during pregnancy and after delivery. Information on the natural history of curable genital infections in pregnancy provides insight into pathophysiology and offers useful scientific guidance to public health efforts in infection screening and control.

Furthermore, the generated data will provide insight on the progression of untreated infections during pregnancy among undiagnosed women and support the need for perinatal screening in order to improve maternal and child health globally.

## 2. Materials and Methods

Vaginal samples and health data were collected by healthcare officials in the context of an established biobanking effort (AMANHI) that started in 2014 [29]. The overall aim of the biobank was to collect detailed epidemiological and biological data of pregnant women and their newborns in order to improve advancing knowledge on key pregnancy and birth outcomes. As previously reported, vaginal swabs were collected under the supervision of a healthcare worker in healthcare facilities and stored in 1 mL eNAT buffer (Copan, Italy) at −20 °C at the Public Health Laboratory Ivo de Carneri on Pemba Island [31]. The participants were informed in their native language about the study and its procedures [29]. All of the included women gave consent to participate in the biobanking effort and the further research linked to it, and study ethics approval was obtained from the Zanzibar Medical Research and Ethics Committee (ZAMREC) [29]. Samples were transported in dry ice to the Netherlands and stored at −20 °C. In total, 686 vaginal samples from 441 women were included for analysis in this study. Of these, 385 samples were collected at first timepoint during pregnancy (between 8 and 20 gestational age (GA) weeks), 257 samples at second timepoint during pregnancy (between 20 and 40 GA weeks), and 44 samples post-delivery (between 42 and 60 days after birth). All vaginal samples were retrospectively tested for the presence of *C. trachomatis*, *N. gonorrhoeae, T. vaginalis* and *M. genitalium* and the positive results were immediately shared with the stakeholders in Tanzania so the participants could be informed of their health status for further action. Details on sample processing and on the genital infections molecular assays have been previously described elsewhere [31,32,33,34,35]. In short, DNA of *C. trachomatis*, *N. gonorrhoeae*, and *T. vaginalis* were detected by their respective CE-IVD-certified Presto *C. trachomatis*, *N. gonorrhoeae* (Goffin Molecular Diagnostics, Houten, The Netherlands) and Presto *T. vaginalis* (Goffin Molecular Diagnostics, Houten, The Netherlands) tests and Real-Time polymerase chain reaction (PCR) with ABI Taqman 7500 (Applied Biosystems, Foster City, CA, USA) as per the manufacturer’s instructions [33,34]. Moreover, a *M. genitalium* assay, as described in Muller et al., was used on the LightCycler 480 II PCR machine (Roche Diagnostics, Basel, Switzerland) [31,32,33,34,35]. Samples were marked positive if the crossing point value (Cp) was between 11 and 38 samples or if samples had a Cp value of 39–40 with an S-shape amplification curve.

Due to logistical and practical constraints, not all women were consecutively sampled at all timepoints. Longitudinal analysis was performed on vaginal samples collected from the same women at more than one timepoint. Persistence and clearance of infection were investigated if samples of women who tested positive at a specific timepoint were also collected at later timepoints. Infection persistence was defined as two or more consecutively sampled vaginal swabs testing positive for the same genital pathogen using the PCR tests mentioned above. Clearance was defined by a subsequent negative result on a sample which had previously tested positive for the same pathogen, again using the PCR tests mentioned above. Demographic and health data such as maternal age, parity, gravidity, school attendance years, religion, previous diagnosis with human immunodeficiency virus (HIV), current symptoms of urinary tract infection and antibiotics use were self-reported via a questionnaire at each sample collection visit. Even though the questionnaire was filled at each visit by all women included in the study, it is important to note that not every questionnaire item might have been answered. Therefore, the total number of responders at a specific demographic and health questionnaire item was explicitly mentioned whenever the total number of available demographic and health data did not match the total number of samples at a specific timepoint.

The frequency and descriptive analysis of the demographic and health data retrieved from the biobank questionnaire were analyzed using IBM SPSS statistical software v. 26 (SPSS Inc., Chicago, IL, USA). Chi-square test with Yates continuity correction was used to determine whether the point prevalence of different infections significantly differed at each collection timepoint. A *p*-value of less than 0.05 was considered statistically significant.

## 3. Results

### 3.1. Population Characteristics

Vaginal samples of 441 pregnant women from Pemba Island were included in this study. The mean maternal age at first sampling was 28.3 years (range 16–48), the mean parity 3.5 (0–10), and 60 (14%) women were primigravida. Mean gravidity was 4.6 (*n* = 436; range 1–16), and only one woman self-reported having HIV infection (*n* = 433; 0.23%). The mean school attendance was two years (range 0–5). A significant majority of subjects identified as (*n* = 436) Muslim Shirazi (99.8%), while one woman identified as Christian-Shirazi.

### 3.2. Sexually Transmitted Infections Prevalence and Symptomatology

In this cohort, the STI burden was 12.2% among the 385 vaginal samples tested at the first timepoint. Twenty-five (6.5%) vaginal samples tested positive for *T. vaginalis* infection, seventeen (4.4%) for *C. trachomatis* infection, five (1.3%) for *M. genitalium* infection and none (0%) for *N. gonorrhoeae* infection (Figure 1a). At this timepoint, twelve of the 325 women (3.7%) who filled out the questionnaire item reported that they had urinary tract infection symptoms. The vaginal sample of one of them tested positive for *C. trachomatis* infection, and the vaginal sample of another for *T. vaginalis* infection. One woman (of the 357 women who filled in the questionnaire item) reported having urinary tract infection symptoms and admitted to taking antibiotics (unknown indication). Eight other women also reported antibiotics use. None of the nine women who reported antibiotics usage (2.5%) tested positive for a genital infection.

At the second timepoint, the STI burden was 16.7% in the 257 vaginal samples tested. Nineteen of the 257 (7.4%) vaginal samples tested positive for *T. vaginalis* infection, fifteen (5.8%) for *C. trachomatis* infection, eight (3.1%) for *M. genitalium* infection and one (0.4%) for *N. gonorrhoeae* infection (Figure 1b). One respondent out of 255 (0.4%) for the questionnaire item reported urinary tract infection symptoms. Her vaginal sample was negative for all tested infections. Nine of the 248 respondents (3.6%) self-reported antibiotics use between the first and second timepoint in pregnancy. The vaginal sample of one of them was positive for *T. vaginalis* infection. The type and exact time of antibiotics use was not available.

Of the 44 vaginal samples tested post-delivery, three (6.8%) were positive for *T. vaginalis* infection, one (2.3%) for *C. trachomatis* infection, one (2.3%) for *N. gonorrhoeae* infection, and none (0%) for *M. genitalium* infection (Figure 1c). The total STI burden was 11.4%. Post-delivery, none of the 37 respondents reported symptoms of urinary tract infections (100%), and five women self-reported antibiotic use (13.5%) (of which none tested positive for a genital infection).

Prevalence of different infections at each timepoint did not significantly differ.

### 3.3. Sexually Transmitted Infections Persistence and Clearance during Pregnancy

In total, 202 women were sampled at first and second timepoints. Among them, eleven (5.4%) tested positive for *C. trachomatis* infection at the first timepoint, eleven (5.4%) for *T. vaginalis* infection, four (3.1%) for *M. genitalium* infection, and none for *N. gonorrhoeae* infection (Figure 2). Of the 202 participants, persistence of *C. trachomatis* infection was 100% (*n*_total_ = 11; average time between sampling collection was 8^+2^ weeks^(+days)^), 82% for *T. vaginalis* infection (*n*_total_ = 11; average time between sampling collection was 9^+6^ weeks) and 75% for *M. genitalium* infection (*n*_total_ = 4; average time between sampling collection was 9^+6^ weeks) (Figure 2). Clearance of *T. vaginalis* infection during pregnancy (*n* = 2) happened within 13^+0^ and 21^+5^ weeks, respectively (Figure 2). Where clearance of *M. genitalium* infection occurred during pregnancy (*n* = 1), it was within 12^+6^ weeks (Figure 2). The vaginal samples of seven women tested positive for a genital infection at the second timepoint, while their samples at the first timepoint were negative (Figure 2).

Nine women self-reported symptoms of urinary tract infection at the first timepoint during pregnancy, but only one of them tested positive for *C. trachomatis* infection, and remained so at the second timepoint (despite being asymptomatic at second testing). None of these women reported antibiotic use during pregnancy. Furthermore, six women reported use of antibiotics at the first timepoint, none of these tested positive for a genital infection at first timepoint; one subsequently tested positive for *C. trachomatis* infection at the second timepoint. At the second timepoint, six women reported antibiotics use, with only one testing positive for *T. vaginalis* infection at the same timepoint. No information on the indication and type for antibiotics used was available.

### 3.4. Sexually Transmitted Infections Persistence and Clearance after Pregnancy

Thirty-eight women were sampled at the second timepoint during pregnancy and post-delivery. Of these women, one of the vaginal samples (2.6%) was positive for *C. trachomatis* infection, five (13.2%) for *T. vaginalis* infection, and none for *M. genitalium* or *N. gonorrhoeae* infections at the second timepoint (Figure 3). Twenty-two weeks after the second timepoint, the persistence rate post-delivery was 100% for *C. trachomatis* (*n*_total_ = 1) and 20% for *T. vaginalis* infections (*n*_total_ = 5) (Figure 3). Four women had post-delivery samples that tested negative for *T. vaginalis* after an average of 18^+6^ (range 15^+1^–22^+0^) weeks of having tested positive (Figure 3). In the group of 38 women, one vaginal sample with *N. gonorrhoeae* and another with *T. vaginalis* were newly detected post-delivery (Figure 3). None of these 38 women self-reported symptoms of urinary tract infection. Six women self-reported use of antibiotics at the second timepoint during pregnancy, and no genital infection was detected at second timepoint during pregnancy and post-delivery in their vaginal samples.

## 4. Discussion

The study observed the persistence of genital infections during pregnancy (*C. trachomatis*, *T. vaginalis* and *M. genitalium*) and post-delivery (*C. trachomatis* and *T. vaginalis*). During pregnancy, *T. vaginalis* infection clearance (26 and 33 weeks after first collection) occurred in two (18%) of the cases, and *M. genitalium* infection clearance (27 weeks after first collection) in one case (25%); *C. trachomatis* infection did not clear in any of the eleven cases. Pre- and post-delivery, *T. vaginalis* infection cleared in one (80%) case (an average of 18 weeks after second collection), while there was no clearance of *C. trachomatis* infection 22 weeks after the second collection. For *N. gonorrhoeae*, the persistence of infection could not be determined.

To our knowledge, this is the first study to investigate the natural history of *T. vaginalis and M. genitalium* during pregnancy and post-delivery. In a non-pregnant cohort of 119 *M. genitalium*-infected sex workers in Kenya (prevalence 14%), 45% spontaneously cleared the infection over the course of three months, whereas in this study (a much smaller *M. genitalium* infected cohort (*n* = 4)), that was the case for just 25% over the course of almost 13 weeks [18]. However, the intervals between sample collection points were not uniform in this study. The sample of the woman who cleared *M. genitalium* infection was collected almost 13 weeks after the first pregnancy collection timepoint, while there was a shorter interval between initial detection and the second collection timepoint (8–10 weeks) for the other three women with persistent infection. As expected, the wider the time interval between collection timepoints, the greater the chance of clearance [17,18].

Existing results from a non-pregnant adolescent (14–17 years) cohort in the United States of America indicate that untreated *T. vaginalis* infection can persist for up to 12 weeks, and that individuals can be asymptomatic for the entire duration of infection [20]. Modelling based on data from non-pregnant women show that *T. vaginalis* persistence might be longer (3–5 years) in older women, and might also explain the prevalence of *T. vaginalis* is this group [19]. In this study, the mean maternal age of the cohort was 28.3 years; thus, the results might be less comparable with the findings in the American adolescent cohort, and more comparable with the longer persistence rate observed in the modelling [15]. *T. vaginalis* infection also cleared more often in women whose samples were collected at larger time intervals (Figure 2 and Figure 3). Although the exact date of infection is unknown, in this small cohort, persistence of *T. vaginalis* infection was observed in 82% of women (average time interval of 10 weeks) during pregnancy, and 20% post-delivery (average time interval of 22 weeks) and clearance was observed after a time interval of 17 weeks and 19 weeks during and after pregnancy, respectively.

Furthermore, this study shows a persistence of *C. trachomatis* infection for up to 13 weeks during pregnancy. This *C. trachomatis* persistence (100%) differs from that reported by Sheffield et al. (pregnant cohort) and other non-pregnant data (retrieved from Golden et al. and Geisler et al.) (Figure 4) [14,17,36]., where spontaneous resolution of *C. trachomatis* occurred in 44% of 1521 pregnant North-American women in the course of 3 months [17]. The discrepancy can be accounted for the lower prevalence of *C. trachomatis* in the present study (5.4%) compared to Sheffield et al. (9%) and the limited number of *C. trachomatis* positive samples at the first timepoint (*n* = 8) in this study compared to other studies (Figure 4) [17].

Dysbiotic vaginal microbiota has been linked with increased susceptibility to genital infections [37,38]. It will be of further interest to investigate how host genetic determinant of infections influence clearance of *C. trachomatis* or other genital pathogens, in particular during pregnancy [24,25,37,38,39,40,41,42,43]. It was hypothesized that the persistence level of genital infections would be higher among pregnant women, since the women’s innate and acquired immune systems are altered [44] to prevent the rejection of a semi-allogeneic foetus, while protecting both mother and foetus from infection [44]. The macrophages in the maternal decidual epithelium (part of innate immunity) have the ability to remodel tissue, supress maternal immune response and present antigens [45,46]. While changes in acquired immunity also occur in pregnancy, there is a decrease in circulating levels of pro-inflammatory cytokines and interleukin-10, the amount of regulatory T-cells is increased in the decidua, maternal immunity shifts towards a T helper (Th) 2 cells phenotype, and Th2 cells outnumber Th1 cells in the decidua [47,48,49]. The mechanisms of immune cell interactions in pregnancy have been extensively reviewed elsewhere and are out of the scope of this paper [24,44,50,51,52,53]. Nevertheless, because of these immunological changes, data about infection persistence and clearance observed in non-pregnant women are not fully representative of pregnant women [54,55]. Our findings show that the clearance might be lower in pregnant women than in non-pregnant cohorts (Figure 4). As discussed, due to the low number of samples tested, more pregnancy-related information is warranted to confirm this trend (Figure 4). The presence of *C. trachomatis* DNA, but not *M. genitalium*, has been detected in chorionic villi from spontaneous miscarriage [56]. Thus, it is not only important to further investigate the interplay between genital infections and host immunity during pregnancy, but also the burden genital infections potentially have on mother and foetus (such as intra-uterine infections, low birthweight, preterm birth, and respiratory infections) [44].

The drawbacks of this study are the relatively small sample size and the limited longitudinal sampling, which, combined with the prevalence of infections, have provided results of a limited power, therefore cautious extrapolation is recommended.

Furthermore, information about general antibiotic use in Pemba is limited and for this study has been obtained via a questionnaire, which lacked details on the indication for, and type of antibiotic used; samples from women using antibiotics were not excluded from the analysis. The relationship between antibiotic use in the preceding months (not directly related to an STI) and lower prevalence of STIs is still debatable even though it might have an effect on *C. trachomatis* prevalence also in countries with low per capita antibiotics consumption [57,58]. However, it should be noted that a large part of background antibiotics use are not first-line treatments for STIs, such as *C. trachomatis* (azithromycin), *M. genitalium* (doxycycline or azithromycin), or *N. gonorrhoeae* (ceftriaxone), and might not have an impact on their course as they are infrequently used for incidental treatment [57]. In relation to this study, detailed information on genital symptoms was also missing. Only one woman self-reported urinary tract infection symptoms, which are often hard to differentiate from genital ones [31,59]. Lastly, there was no differentiation between persistence of infections and re-infection. It is therefore possible that some women achieved clearance from a specified pathogen, and were re-infected before the subsequent collection timepoint. To mitigate against this phenomenon, pathogen genotyping testing should be considered, especially in a larger cohort analysis [18]. Bacterial load and co-infections with other burdensome pathogens, for example HIV, vaginal dysbiosis and chronic inflammation should also be considered to better understand the clinical implications of these infections [18]. However, in this cohort the number of women who self-reported diagnosis of HIV was very low.

Current understanding of the natural history of *C. trachomatis*, *T. vaginalis,*
*N. gonorrhoeae* and *M. genitalium* is hampered by a lack of robust scientific studies; it is mostly limited to that provided by low sensitivity and specificity studies (using culture-based methods) in the pre-antibiotic era [16,17,20]. Logically, with current antibiotic treatment possibilities, ethical constraints are major reasons for the lack of studies investigating the natural history of curable pathogens. Nonetheless, and as shown with this retrospective approach, information about the natural history of infection in a specific population can still be obtained with the help of existing biobank samples and database. Retrieving this information remains important, as evidence like the one provided here is essential to public health officials when evaluating the burden and influence of genital infections on maternal and neonatal health. Currently, most genital screening is not implemented in antenatal care in a standard way, or only implemented early in pregnancy [60,61]. Furthermore, the management of these STIs is in most countries syndromic, while most infected women are asymptomatic undiagnosed carriers, a situation that puts them and the foetus at risk [62].

Thus, the observation that there is not always natural clearance of these curable pathogens during the pregnancy, and the possibility of de novo infections, shows remaining challenges in detecting and treating *C. trachomatis*, *T. vaginalis,*
*N. gonorrhoeae* and *M. genitalium* during pregnancy [60,61,63,64]. The urgency of better screening practices is ever more pertinent, particularly as new resistant pathogen strains arise, as is the case for *N. gonorrhoeae*, *C. trachomatis*, *T. vaginalis,* and *M. genitalium* [65,66,67,68,69]. Similarly, understanding of the natural progression of these infections would be beneficial for vaccine design against these microorganisms [16,64].

## 5. Conclusions

This study reports the persistence rate of *C. trachomatis* (100%), *T. vaginalis* (82%) and *M. genitalium* (75%) infections during pregnancy and *C. trachomatis* (100%) and *T. vaginalis* (20%) before and post-delivery in vaginal samples from pregnant women living in Pemba Island, Tanzania. The persistence of *N. gonorrhoeae* during pregnancy could not be determined because of low prevalence of infection. The detection and persistence of these curable genital infection stresses on the importance of screening all women during the pre and postnatal periods, where treatment will decrease sequelae. Investigating the clearance and persistence rate of these infections will improve the knowledge of the pathophysiology of these genital infections during pregnancy and post-delivery. With a retrospective approach, larger biobank cohorts can build on this data and identify factors associated with duration or clearance of infection during pregnancy. Such evidence will support effective management and screening programs, as well as innovative biomedical approaches targeting them.

## Figures and Tables

**Figure 1 microorganisms-09-01180-f001:**
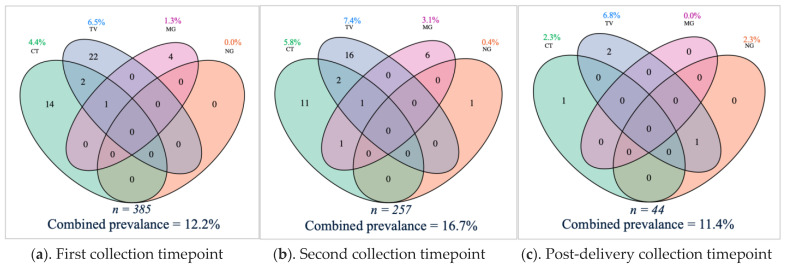
Venn diagrams with point prevalence and combined prevalence of *Chlamydia trachomatis* (CT), *Neisseria gonorrhoeae* (NG), *Trichomonas vaginalis* (TV) and *Mycoplasma genitalium* (MG) genital infections during and after pregnancy. The number represents the amount (*n*) of samples with a detected microorganism. Above is the individual point prevalence per genital infection and below each diagram is the total number of tested samples and the combined point prevalence of genital infections. (**a**) the sexually transmitted infection (STI) burden was 12.2% among the 385 vaginal samples tested at the first timepoint. (**b**) 16.7% among the 257 vaginal samples tested positive for at least one STI at the first timepoint. (**c**) The total STI burden post-delivery was 11.4% among 44 samples tested.

**Figure 2 microorganisms-09-01180-f002:**
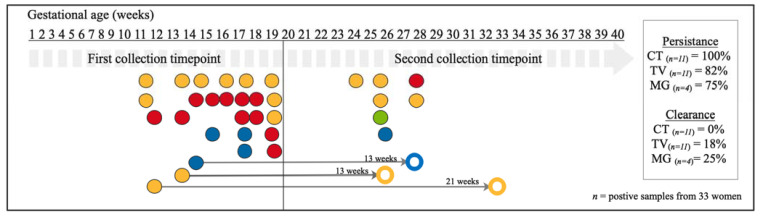
Schematic overview of the persistence and clearance of CT, NG, TV and MG genital infections of positive vaginal swabs collected during pregnancy (*n* = 33 infected women out of 202 tested women). Horizontal labels indicate the time of collection in gestational age in weeks. Each dot represents a vaginal sample positive for *C. trachomatis* (in red), *T. vaginalis* (in yellow), *N. gonorrhoeae* (in green), and *M. genitalium* (in blue). Open dots represent cleared infections. The thin grey arrows, with the interval between testing in weeks on top, connect a positive sample with matching coloured open dot and it refers to cleared infections. The seven dots in the second collection timepoint period indicate samples with de novo infections (as samples from the same women collected earlier were not positive for the same pathogens). All infections detected at first timepoint did persist if no thin grey arrow was connected to them.

**Figure 3 microorganisms-09-01180-f003:**
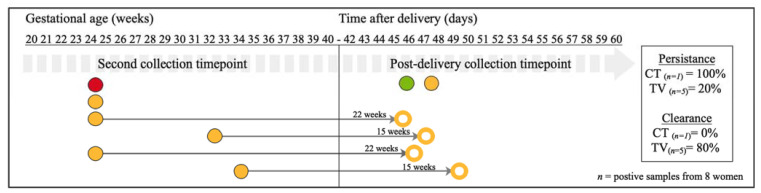
Schematic overview of the persistence and clearance of CT, NG, TV and MG genital infections of positive vaginal swabs collected during pregnancy and post-delivery (*n* = 8 infected women out of 38 tested women). Horizontal labels indicate the time of collection in gestational age in weeks. Each dot represents a vaginal sample positive for *C. trachomatis* (in red), *T. vaginalis* (in yellow), *N. gonorrhoeae* (in green), and *M. genitalium* (in blue). *M. genitalium* was not detected at the second timepoint in this sub-cohort. Open dots represent cleared infections. The thin grey arrows, with the interval between testing in weeks on top, connect a positive sample with a matching coloured open dot and it refers to cleared infections. The two dots (*T. vaginalis* (*n* = 1) and *N. gonorrhoeae* (*n* = 1)) in the post-delivery collection timepoint period indicate samples with de novo infections (as samples from the same women collected at the second timepoint were not positive for the same pathogens). All infections detected at second timepoint did persist if no thin grey arrow was connected to them.

**Figure 4 microorganisms-09-01180-f004:**
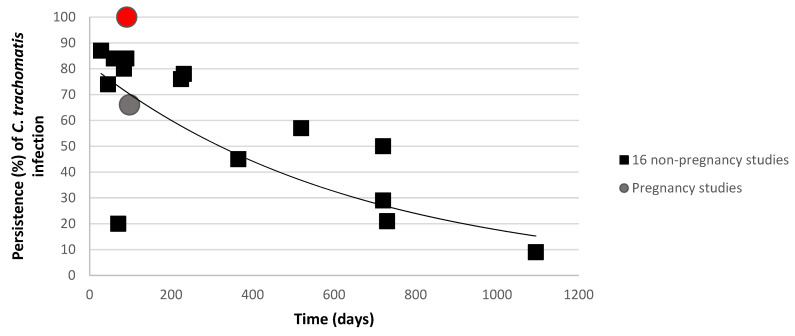
Urogenital *C. trachomatis* presence in women during follow-up. Percentages of *C. trachomatis* infected women are described by Golden et al. and Geisler et al.

 are compared with two pregnancy studies (Sheffield et al. 
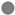
 and the present study 
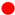
) [14,17,36]. The curve shows the trend as calculated by the sixteen non-pregnant studies.

## Data Availability

Data are available from the authors with the permission of the Biobank governing body/local institution and the Principal Investigator of the site.

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
