# Peer review of "The Natural Course of Chlamydia trachomatis, Neisseria gonorrhoeae, Trichomonas vaginalis, and Mycoplasma genitalium in Pregnant and Post-Delivery Women in Pemba Island, Tanzania"

_microorganisms, 2021, doi:10.3390/microorganisms9061180_

Round 1

Reviewer 1 Report

The manuscript entitled “The natural course of Chlamydia trachomatis, Neisseria gonorrhoeae,

Trichomonas vaginalis, and Mycoplasma genitalium in pregnant and post-delivery women in Pemba Island, Tanzania.” by Juliana and colleagues determined the presence DNA belonging to Chlamydia trachomatis, Neisseria gonorrhoeae, Trichomonas vaginalis and Mycoplasma genitalium in a total of 441 vaginal swabs of Tanzanian pregnant females and females after delivery. Overall, this topic is quite important and interesting. This work can help to better understand the role of pathogenic infections as during pregnancy. In my opinion, it can be accepted after major revision.

Limitations.

The methods employed should be clearly described for a better understanding of the experimental design. In addition, there is an imbalance between sections, being the methods very short and undetailed, while the discussion is too verbose

Strengths

Novel and interesting findings presented

  1. Although methods have previously been reported in refs 33-37 a description in a novel paragraph should be helpful for the reader. Reading throughout the manuscript, it is unclear the methods employed. PCR? Sequencing? In addition, what does the authors mean for “persistence of genital infections during pregnancy” in the discussion section or “vaginal samples tested positive” in the results section? presence of DNA? Proteins? mRNA? What genes? This information should be detailed clearly throughout the manuscript. These sentences are quite vague.
  2. The presence of bacterial DNA belonging to Chlamydia trachomatis and Mycoplasma genitalium has been previously qualitatively and quantitatively investigated in chorionic villi from spontaneous affected females and females underwent voluntary interruption of pregnancy (PMID: 30078192). DNA belonging to Chlamydia trachomatis has been detected in 3% and 4% of spontaneous affected females and females underwent voluntary interruption of pregnancy, respectively. None of the samples were positive for Mycoplasma genitalium DNA. This is an important information that should be included in the manuscript along with the supporting reference.
  3. Pathogens rates should be statistically compared. For instance, it seems that T. vaginalis resulted the most prevalent among analyzed females.
  4. There is an imbalance between sections, being the methods very short and undetailed, while the discussion is too verbose. I would suggest to reduce the discussion by 20% and include more details about the methods

Minor points

STI should be included as sexually transmitted infection (STI) when quoted the first time

“To our knowledge, this is the first study to observe the natural history” à observing

Author Response

The manuscript entitled “The natural course of Chlamydia trachomatis, Neisseria gonorrhoeae,

Trichomonas vaginalis, and Mycoplasma genitalium in pregnant and post-delivery women in Pemba Island, Tanzania.” by Juliana and colleagues determined the presence DNA belonging to Chlamydia trachomatis, Neisseria gonorrhoeae, Trichomonas vaginalis and Mycoplasma genitalium in a total of 441 vaginal swabs of Tanzanian pregnant females and females after delivery. Overall, this topic is quite important and interesting. This work can help to better understand the role of pathogenic infections as during pregnancy. In my opinion, it can be accepted after major revision.

Limitations.

The methods employed should be clearly described for a better understanding of the experimental design. In addition, there is an imbalance between sections, being the methods very short and undetailed, while the discussion is too verbose

These comments have been addressed in the response to point 4

Strengths

Novel and interesting findings presented

Thank you, we agree with this comment.

  1. Although methods have previously been reported in refs 33-37 a description in a novel paragraph should be helpful for the reader. Reading throughout the manuscript, it is unclear the methods employed. PCR? Sequencing? In addition, what does the authors mean for “persistence of genital infections during pregnancy” in the discussion section or “vaginal samples tested positive” in the results section? presence of DNA? Proteins? mRNA? What genes? This information should be detailed clearly throughout the manuscript. These sentences are quite vague.

Thank you for this comment; we have included information on tests and what we refer to as positive tests in the methods section. (Lines 115-124)

  1. The presence of bacterial DNA belonging to Chlamydia trachomatis and Mycoplasma genitalium has been previously qualitatively and quantitatively investigated in chorionic villi from spontaneous affected females and females underwent voluntary interruption of pregnancy (PMID: 30078192). DNA belonging to Chlamydia trachomatis has been detected in 3% and 4% of spontaneous affected females and females underwent voluntary interruption of pregnancy, respectively. None of the samples were positive for Mycoplasma genitalium DNA. This is an important information that should be included in the manuscript along with the supporting reference.

Thank you for this suggestion. This evidence has been included in the discussion and references in the reference list. Lines 315-317.

  1. Pathogens rates should be statistically compared. For instance, it seems that T. vaginalis resulted the most prevalent among analyzed females.

Thank you for this suggestion. The methods used for the calculations to answer this question are written in line 144-147. The results of the analyzations are mentioned in line 186.

  1. There is an imbalance between sections, being the methods very short and undetailed, while the discussion is too verbose. I would suggest to reduce the discussion by 20% and include more details about the methods

Thank you for this comment.

As suggested, the discussion has now been shortened to around 80% of its original length, while more details have been included in the methods section

Minor points

STI should be included as sexually transmitted infection (STI) when quoted the first time

This acronym is defined at the very beginning of the introduction. Line 36.

“To our knowledge, this is the first study to observe the natural history” à observing

This has been revised to “To our knowledge, this is the first study to investigate…” Line 262.

Reviewer 2 Report

In this article, Juliana and colleagues reported the results of a study conducted on vaginal samples collected from women during pregnancy and post-delivery and stored in a biobank.

The aim of this retrospective study was to evaluate the prevalence and persistence of 4 sexually transmitted infections (STIs), C. trachomatis, N. gonorrhoeae, T. vaginalis and M. genitalium, to better understand the natural course of these infections in pregnant and post-delivery women.

It is important to well define the course of STIs infection to improve national prevention programs, especially in this group of patients.

Even if just a few results regarding the persistence of investigated STIs are described, this article is well described and interesting data are reported. A study of this type must be predominantly descriptive but the goal of the study is clear and methods used are adequate. Anyway, there are minor issues to arrange.

The use of the English language in adequate.

Material and Methods:

Authors reported: “As previously reported, vaginal swabs were collected in eNAT buffer (Copan, Italy) under the supervision of a healthcare worker in healthcare facilities”

Authors should better specify in which volume of eNAT buffer these swabs was resuspended and the storage condition of these samples before testing.

Moreover, they should describe the test used for the STIs detection in the text.

Results:

In the section 3.3, authors should add the number of women with persistent infections next to the percentage. Numbers regarding women with persistent or cleared infection should be reports also in Figures 2 and 3.

Author Response

In this article, Juliana and colleagues reported the results of a study conducted on vaginal samples collected from women during pregnancy and post-delivery and stored in a biobank.

The aim of this retrospective study was to evaluate the prevalence and persistence of 4 sexually transmitted infections (STIs), C. trachomatis, N. gonorrhoeae, T. vaginalis and M. genitalium, to better understand the natural course of these infections in pregnant and post-delivery women.

It is important to well define the course of STIs infection to improve national prevention programs, especially in this group of patients.

Even if just a few results regarding the persistence of investigated STIs are described, this article is well described and interesting data are reported. A study of this type must be predominantly descriptive but the goal of the study is clear and methods used are adequate. Anyway, there are minor issues to arrange.

The use of the English language in adequate.

Thank you for these positive comments

Material and Methods:

Authors reported: “As previously reported, vaginal swabs were collected in eNAT buffer (Copan, Italy) under the supervision of a healthcare worker in healthcare facilities”

Authors should better specify in which volume of eNAT buffer these swabs was resuspended and the storage condition of these samples before testing.

Moreover, they should describe the test used for the STIs detection in the text.

Thank you for these comments. We have revised the methods to include details on sample preservation and testing (lines 99-106)

Results:

In the section 3.3, authors should add the number of women with persistent infections next to the percentage. Numbers regarding women with persistent or cleared infection should be reports also in Figures 2 and 3.

Thank you for this comment, this information has been added across the results and figures as suggested.

Round 2

Reviewer 1 Report

The authors have addressed all observations. The ms can be accepted.

I have only two minor comments

line 98 is "advance” a repetition?

Lines 324-326 English should be revised